

# Inefficacy of mallard flight responses to approaching vehicles

Shane Guenin[1,2], Carson J. Pakula[1,2], Jonathon Skaggs[1],
Esteban Fernández-Juricic[3] and Travis L. DeVault[1]

[1] Savannah River Ecology Laboratory, University of Georgia, Aiken, South Carolina, United States
[2] Warnell School of Forestry & Natural Resources, University of Georgia, Athens, Georgia, United States
[3] Department of Biological Sciences, Purdue University, West Lafayette, Indiana, United States

## ABSTRACT

Vehicle collisions with birds are financially costly and dangerous to humans and animals. To reduce collisions, it is necessary to understand how birds respond to approaching vehicles. We used simulated (*i.e.*, animals exposed to video playback) and real vehicle approaches with mallards (*Anas platyrynchos*) to quantify flight behavior and probability of collision under different vehicle speeds and times of day (day *vs.* night). Birds exposed to simulated nighttime approaches exhibited reduced probability of attempting escape, but when escape was attempted, fled with more time before collision compared to birds exposed to simulated daytime approaches. The lower probability of flight may indicate that the visual stimulus of vehicle approaches at night (*i.e.*, looming headlights) is perceived as less threatening than when the full vehicle is more visible during the day; alternatively, the mallard visual system might be incompatible with vehicle lighting in dark settings. Mallards approached by a real vehicle exhibited a delayed margin of safety (both flight initiation distance and time before collision decreased with speed); they are the first bird species found to exhibit this response to vehicle approach. Our findings suggest mallards are poorly equipped to adequately respond to fast-moving vehicles and demonstrate the need for continued research into methods promoting effective avian avoidance behaviors.

Corresponding author
Shane Guenin, guenins@purdue.edu

## BACKGROUND

In the past century, humans have developed terrestrial, aquatic, and aerial vehicles that move at speeds outpacing the fastest animal predators. Many animals are susceptible to being struck by high-speed vehicles (*Loss, Will & Marra, 2014*; *DeVault et al., 2014*; *Lima et al., 2015*), and the mechanisms governing animal responses to vehicles are poorly understood (*Lima et al., 2015*). Animal-vehicle collisions (AVCs) are especially concerning for human safety when they involve collisions between animals (usually birds) and aircraft. During the last 30 years, bird strikes were responsible for 464 human fatalities (*Dolbeer et al., 2023*) and are estimated to cause an average of $205 million in damage to U.S. civil aircraft annually (*Dolbeer et al., 2021*). Although bird strikes at low altitudes have

decreased in recent years due to intensive wildlife management at airports (*Dolbeer et al., 2014*), the rate of damaging collisions outside airports is increasing (*Dolbeer, 2011*; *DeVault et al., 2016*).

There is a clear need to better understand how birds perceive and respond to oncoming vehicles (*Lima et al., 2015*), which could inform measures to reduce the probability of dangerous and costly strikes. There are two main hypotheses that have been proposed to explain why various taxa sometimes fail to evade vehicles (*Lima et al., 2015*). First, animals might not perceive a vehicle as risky until a collision is inevitable, and second, animals might not initiate an evasive behavior in time to avoid the vehicle, even when the vehicle is perceived as a deadly threat (*Blackwell et al., 2019*, *2020*). These hypotheses mostly rely on the principles of antipredator behavior.

According to antipredator theory, animals assess perceived risk based on associated costs and benefits (*Ydenberg & Dill, 1986*). Prey animals should make decisions that maximize fitness by reducing the likelihood of predation (*Cooper & Frederick, 2007*) but might choose to delay or even avoid using antipredator response behaviors in favor of other responses relative to the magnitude of perceived risk (*Helfman, 1989*). Animal responses to anthropogenic stimuli like vehicles are expected to follow principles like those followed when prey encounter predators (*Frid & Dill, 2002*). However, recent research by *Lunn et al. (2022)* found that antipredator theory is limited in terms of predicting animal responses to vehicles, which emphasizes the need for novel theoretical frameworks as vehicles differ from predators in several ways, including size, speed, and directness of approach (*Lima et al., 2015*). Vehicles also lack visual cues animals would normally use to identify predators (*i.e.*, eye gaze, pursuit; *Blackwell, Seamans & DeVault, 2014*; *Lima et al., 2015*). To develop these new models, it is necessary to gain a deeper understanding of the escape responses animals follow when exposed to high-speed vehicles.

Three potential escape responses an animal could display after alerting to a perceived oncoming threat involve temporal, spatial, or delayed margins of safety (*Cárdenas et al., 2005*; *DeVault et al., 2015*; *Lunn et al., 2022*). These margins of safety describe the pattern of an animal's flight initiation distance (the distance between the animal and the oncoming threat at the onset of the flight response; FID) as speed increases. Another metric modulated by animals across these three behavioral responses is the time-to-collision (TTC), which describes the estimated amount of time that will elapse before the threat reaches an animal's location (*Wang & Frost, 1992*; *DeVault et al., 2014*). TTC is mathematically related to FID and changes inversely with approach speed; FID (m) = TTC (s) * Approach Speed (m/s). For animals using a temporal margin of safety, FID increases as approach speed of the threat increases as the animal attempts to maintain its flight response at a consistent TTC. Alternatively, when animals respond to an oncoming threat using a spatial margin of safety, FID remains consistent regardless of approach speed; as a result, TTC decreases as the speed of the threat increases. Temporal margins of safety have not previously been observed in bird species (however, see *Legagneux & Ducatez, 2013*) but are seen in Thomson's gazelles (*Eudorcas thomsonii*; *Walther, 1969*), broad-headed skinks (*Emece laticeps*; *Cooper, 1997*), and desert iguanas (*Dipsosaurus dorsalis*; *Cooper, 2003*). Spatial margins of safety have been observed in woodchucks (*Marmota monax*;

*Bonenfant & Kramer, 1996*), galahs (*Cacatua roseicapilla*; *Cárdenas et al., 2005*) and brown-headed cowbirds (*Molothrus ater*; *DeVault et al., 2015*). Lastly, the delayed margin of safety is hypothesized to result from an animal being distracted by other stimuli, allowing faster threats a closer approach to the animal than slower threats before the animal reacts (*Lunn et al., 2022*). While the exact causes of a delayed margin of safety could be any activity with a fixed reaction time (*i.e.*, neural processing, threat assessment, focus on another stimulus), a delayed margin of safety describes any situation wherein FID decreases as approach speed of the threat increases.

To our knowledge, no previous studies have documented a delayed margin of safety in the context of high-speed vehicle approaches; however, see *Schroeder & Panebianco (2021)*, who provide evidence of a delayed margin of safety in guanacos (*Lama guanicoe*) responding to approaching uncrewed aerial vehicles. *Legagneux & Ducatez (2013)* found an interesting behavioral response by birds to approaching automobiles–birds in this study appeared to adjust their FIDs relative to the posted speed limit, rather than the speed an approaching vehicle was traveling. This indicates that birds whose habitat includes roadways may adjust their flight responses to more general environmental factors, rather than the attributes of an individual threat. Although not reflective of a defined margin of safety, other species fail to adjust FIDs for the speed of approaching vehicles, with FID fluctuating erratically as speed increases, and thus are at greater risk of being struck, at least for a subset of vehicle speeds (turkey vultures (*Cathartes aura*; *DeVault et al., 2014*), rock pigeons (*Columba livia*; *DeVault et al., 2017*), and white-tailed deer (*Odocoileus virginianus*; *Blackwell, Seamans & DeVault, 2014*)). Both spatial and delayed margins of safety can be maladaptive in the context of high-speed vehicle approaches. For example, brown-headed cowbirds consistently initiated a flight response approximately 28 m away from an oncoming vehicle, regardless of its speed, which would lead to a high probability of collision once vehicles reached speeds of 120 km/h or higher (*DeVault et al., 2015*).

Conspicuous vehicle lighting has been shown to enhance alert behaviors for several bird species in response to oncoming vehicles (*Blackwell et al., 2009a*, *2012*; *Doppler et al., 2015*), which could affect escape strategies, yet no data exist for nighttime conditions. These data are critical for reducing bird strikes with high-speed vehicles, given that many species generally make long distance migratory movements at night (*Korner et al., 2016*). Bird strike data from the USA indicate that collisions with aircraft are more frequent at dusk and night than during the day, even though there are fewer aircraft flights during these time periods (*United States Federal Aviation Administration, 2024*). Terrestrial vehicles are also hazardous to birds during the night, especially given evidence that migrating birds may be drawn to roadways at night (*La Sorte et al., 2022*) and that automobile headlights may temporarily stun birds in the vehicle's path (*Erritzoe, Mazgajski & Rejt, 2003*).

The goal of this study was to assess avian reactions to approaching vehicles with conspicuous lighting during day and night conditions. More specifically, we investigated whether the margin of safety used by birds differed between day and night. We also evaluated whether individuals would have survived (hereafter, "successful avoidance") a vehicle approach under variations in vehicle speed and time of day (day and night). We

used two complementary experimental approaches: simulated (*i.e.*, video playback) vehicle approaches, which allowed for high experimental speeds unsafe to test in the field (*DeVault et al., 2015*), and field vehicle approaches, which quantified how birds reacted during a genuine vehicle encounter.

Video playback has been used extensively in behavioral experiments due to its high level of replicability and the ability to subject animals to stimuli which would otherwise be unsafe (*D'Eath, 1998*; *Lea & Dittrich, 1999*; *DeVault et al., 2015*). However, stimuli presented in video playback may not visually match their real-world counterparts when viewed through the avian eye; videos may appear to lack the depth or resolution of a real stimulus (*Fleishman & Endler, 2000*), and color screens are designed for trichromatic human vision instead of the tetrachromatic vision of birds (*Cuthill et al., 2000*). Real-world approaches provide a realistic visual stimulus but are less replicable due to any number of confounding meteorological and environmental variables. For our purposes, real approaches also become hazardous for the safety of test subjects and the vehicle operators at high speeds.

This study provides an exploratory attempt at characterizing the responses animals use to avoid high-speed vehicles in different ambient light conditions and could inform mitigation strategies for reducing collisions with vehicles, thus increasing human and animal safety while reducing damage to vehicles.

## METHODS

Mallards (*Anas platyrhynchos*; *DeVault et al., 2011*; *Pfeiffer, Blackwell & DeVault, 2018*) rank among the ten most costly bird species in terms of bird-aircraft collisions (*DeVault et al., 2016*). The mallard is a large and widespread dabbling duck (*Anatinae*) found in high abundance across the northern hemisphere (*Drilling, Titman & McKinney, 2020*). Release and escape of domestic mallards has resulted in wild (or feral) mallard presence on every continent outside of Antarctica (*Baldassarre, 2014*). Waterbird strikes by terrestrial vehicles make up a relatively smaller proportion of avian-vehicle collisions; however, mallards appear as casualties in several datasets of bird strikes on Canadian roads (*Bishop & Brogan, 2013*). We conducted two experiments at the Savannah River Site (SRS), an 803 km$^2$ federal property adjacent to the Savannah River near Aiken, South Carolina, managed by the United States Department of Energy (*Savannah River Site, 2020*). For both experiments, we used a single captive population of domestically-raised, wild-type mallards as a model organism for ducks involved in vehicle encounters. These birds were raised to be released onto hunting preserves; thus, they remained flighted and were reared with minimal human contact. Flighted birds were necessary to represent realistic vehicle approaches because birds with clipped flight feathers (*i.e.*, rendered incapable of flying) might have behaved differently than flighted birds (*Blackwell et al., 2019*). We used 97 mallards in this study (30 female, 67 male). From arrival to release, 77 were kept for 4 months, and 20, which arrived later in the study, were kept for 2 months. This was the maximum number of individuals which could be held at one time in our housing area and provided comparable sample sizes to those used in previous avian-vehicle experiments (*DeVault et al., 2015*; *Blackwell et al., 2019*). While in our care, mallards were housed in an

indoor holding facility grouped in pens by sex with continual access to flowing water, which pooled in a 36 cm trough on one end of the pens. They were fed Purina® duck feed pellets, *ad libitum*.

## Simulated trials

We used video playback to expose mallards to high-speed vehicle approaches in a controlled, safe environment. Video playback is effective in assessing animal response to various stimuli (*D'Eath, 1998*) and has been used in previous studies involving birds (*Lea & Dittrich, 1999*), including those evaluating behavior in response to oncoming vehicles (*DeVault et al., 2015*, *2017*, *2018*). A white 2018 Ford F-150 pickup truck with stock, high-beam, halogen headlamps turned on was used for all vehicle approaches. A 3-m long, 2.5-cm wide, black, steel, square tube was fixed to the top of the cab (approximately 2 m from the ground), and two 4,950 lumen Sunspot 36 LX airplane landing lights (AeroLEDs, Boise, ID) were attached on either end of the bar (Fig. S1). Both the vehicle headlights and landing lights were on for all approaches. This lighting arrangement was chosen to mimic the lighting array of a small passenger aircraft traveling down a runway, following *Blackwell & Bernhardt (2004)*. Although some bird species might initiate flight responses from aircraft noise (*Harris, 2005*; *Lima et al., 2015*), mallards and other dabbling ducks (*Anatinae*) are not typically flushed by low-flying aircraft (*Conomy et al., 1998*). Mallards are likely to rely primarily on vision when responding to high-speed vehicle approaches, and it was important to us to achieve a semi-realistic lighting array.

Vehicle approaches at three speeds (30, 60, and 120 km/h) were recorded in 4k resolution (2,160 × 3,840 pixels) at 30 frames per second (fps) during the day on 27 October 2021 and night on 2 November 2021 under dry, clear conditions (six videos total) using a Sony Handycam model video camera. The speeds were later doubled during video editing to 60 fps to achieve vehicle playback speeds of 60, 120, and 240 km/h to reduce the likelihood of the mallards perceiving flicker in the video during simulated vehicle approaches (*D'Eath, 1998*). The camera was placed on the centerline of the road to record the vehicle approach from the approximate height of a mallard. The vehicle began its approach down a level, straight roadway, 550 m from the camera and was visible to the mallard throughout the duration of the approach. One of the limiting factors in the ability of mallards to detect a vehicle is the distance that their visual system can resolve the vehicle. Using the visual acuity of mallards (12.8 cycles per degree; E. Fernandez-Juricic, 2018, personal observation), we estimated the distance that the vehicle used for the approaches (relative to its 1.9 m width) would be detected at the threshold of resolution (assuming optimal light conditions). We used the formula $d = \frac{r}{\tan\frac{\alpha}{2}}$; where $r$ represents the radius of the object (approaching truck), and $\alpha = \frac{1}{visual\ acuity}$ (for a similar approach see *Tyrrell et al., 2013*), and determined the 550 m approach distance was well within the range of a mallard's visual acuity.

Video-simulated trials were conducted between November 2021–January 2022. In simulated approaches, each individual mallard was shown a single video. Each treatment video ($n = 6$) was played for 16 unique individuals (five female, 11 male), for a total of 96

trials. Treatments were run in groups of six (one of each treatment video) in a randomly selected order. Mallards were caught in a haphazard manner from their holding area, with bias to keep sex consistent across treatments. Following previous simulated vehicle approach methods (*DeVault et al., 2015*), we began a simulated trial by placing an individual mallard in a 108 depth × 157 width × 116 height cm box comprised of a wire mesh floor, plywood ceiling and three walls, and a 2.5 cm mesh front wall separating them from an 83 × 145 cm Samsung RU8000 Series television screen. We cut boards of extruded polystyrene, painted them gray, and installed them between the mesh front wall and television screen to taper the mallard's view at the front of the box so only the television screen was visible. Three cameras (Model HT-5000SC; Emergent Vision Technologies, Port Coquitlam, BC, Canada) recorded mallard responses in the box from the sides and back, and video feeds from each camera were recorded for later analysis and livestreamed to an adjacent room for real-time observation. During all trials (day and night), the box was illuminated from above by two 15-watt, 120 Hz LED bulbs (1,600 lumens) and sealed from all external light. Each mallard was captured in the holding facility using a net, transferred to the video box in a small pet carrier, and given a five-min acclimation period in the box before the vehicle approach video was played. Before the mallard was placed into the video box, the approach video was loaded, and remained paused on the first frame during acclimation, during which the stationary vehicle was visible. Approach videos lasted between 10–30 s after start, depending on vehicle speed. After each trial, individuals were banded before being returned to the holding facility to ensure none were repeated in future trials.

To determine the hypothetical outcome of avoidance responses to simulated vehicle approaches (*i.e.*, collision or successful escape), we calculated the mean time required for mallards to move from the path of the vehicle (*i.e.*, the minimum TTC required for vehicle avoidance) by conducting a field experiment to quantify the time necessary for mallards to travel 3 m (the width of a standard road lane) from a stationary position (*DeVault et al., 2015, 2017*). To do so, we constructed an 8-m-long chute from snow fencing in a 15 × 10 m flight cage. The distance of the chute was marked at 0.5 m intervals. Individual mallards ($n = 20$; 10 male, 10 female) were placed into a net and held above the ground by a researcher in a blind, and once lowered onto the ground and no longer constrained by the net, the researcher jumped from the blind and shouted, prompting an escape response. Using the video recorded trials, we determined the time from flight initiation until the birds reached the 3 m line was 1.0 s ± 0.14 SD.

## Field trials

We conducted a field experiment on an unused road on the Savannah River Site, 15 km from our holding facility, to quantify mallard responses to a real vehicle approach. This roadway was the same one used to film the simulated approach videos. The road corridor was 12 m wide, including two lanes and a grass shoulder, and was heavily forested on each side. Although the road corridor was relatively narrow, we do not believe the wooded edge caused a "tunnel effect", given that many ducks chose to flee into the cover of the trees.

Following previous field vehicle approach methods (*Blackwell et al., 2019*), ducks were released during the field portion of the study, due to (1) the difficult nature of recapturing flighted birds and (2) the necessity of not impeding flight so their responses were not affected by any imposed barrier. During the field experiment, there was a risk of collision. However, we took multiple steps to avoid this possibility including using an experienced driver, a mandatory braking zone, and a passenger observer who monitored the entire encounter on a forward-looking infrared (FLIR) camera (FLIR M625S, FLIR Systems, Goleta, California, USA) to alert the driver of the approximate distance to, and any movement of, the individual mallard. During the field experiment, one individual was struck at a low speed (<10 km/h) and flew away before any evaluation could take place, which led us to believe its injuries, if any, were minor.

Approaches in the field occurred during the day (10:00–14:00) and night (30 min after sunset–23:00), using the same vehicle and aircraft landing light setup described above. The vehicle's approach began 550 m from the mallard's release point and drove down the center of the roadway for the entire approach. One researcher remotely opened a carrier where the focal mallard was being held from behind a blind (Fig. S2) and remained in this position, unmoving, for the duration of the trial. Upon the mallard's release, this researcher relayed to the driver *via* cell phone to begin the vehicle approach. Daytime trials were conducted under clear conditions from 15–17 February 2022. During nighttime trials, we conducted the experiment on clear nights on or around the full moon (>90% illuminance), on 14–16 February 2022. Both day and night field approaches were conducted at 40 and 60 km/h, which were reached at 4 and 6 sec after approach began, respectively. These two speeds (compared to 60, 120, 240 km/h during simulated approaches) were chosen to allow the driver to brake safely prior to any actual collisions and to reduce the overall number of treatments, given that we anticipated some individuals would escape the experimental arena immediately upon release (*i.e.*, before the vehicle approach began). Individuals were selected haphazardly for trials (*i.e.*, first individual caught from holding area) and 24 individuals were used for each treatment.

We measured the truck's braking distances at both speeds prior to approaches with live birds and determined we could stop within 10 m at 40 km/h and 15 m at 60 km/h. We used this information to mark mandatory braking points on the road at 10 and 15 m from the mallards' release point to reduce the chance of striking a live bird. This potential confounding variable, (*i.e.*, braking) was considered during data analysis for approaches when braking occurred (see below).

All vehicle approaches and mallard behaviors were video recorded by two cameras–a Sony Handycam video recorder positioned perpendicular to the release point on the road, and a Canon EOS 77D camera positioned 15 m from the release point, facing along the roadway to record the entire approach (Fig. S2).

## Analyses

### Response variables: FID & TTC

For both experiments, we extracted FID and TTC values from the videos, following previous methods (*DeVault et al., 2015*; *Blackwell et al., 2019*; Fig. 1), and corrected for
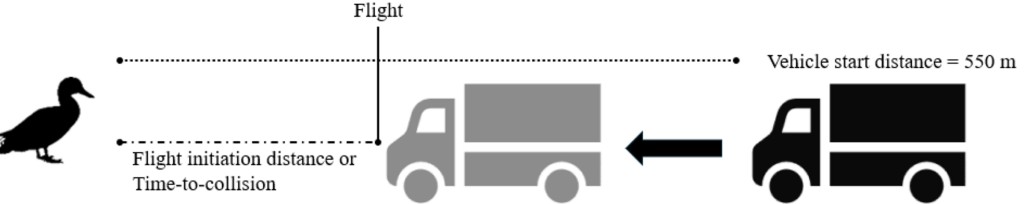

**Figure 1 Metrics of flight initiation distance (FID) and time-to-collision (TTC) illustrated on a schematic of an experimental vehicle approach.** FID and TTC are mathematically related (FID = TTC * Approach Speed) and are therefore equally represented in space. Mallard source credit: Maija Kerala, CC-BY-NC-SA, https://www.phylopic.org/images/3ceaa22b-8879-4545-9e32-425010f33cd4/anas-platyrhynchos. All other components are from Microsoft PowerPoint.

duck movement after release when necessary in the field. These corrections for an individual's FID only accounted for forward or backward movement along the roadway (*i.e.*, movement closer or farther away from the vehicle) during the approach. A flight response was defined as a mallard's first movement which began a fluid process of locomotion in a single direction in an attempt to exit the vehicle's path of travel. Birds walking in the roadway which paused their movement to assess the vehicle or ceased movement entirely were not considered to be fleeing. Both FID and TTC metrics were calculated by using the time in seconds before the vehicle eclipsed the mallard's position before flight–the TTC value–and was multiplied by the speed of the vehicle (m/s) to find FID. For all models, only individuals which displayed a flight response were included. Unlike similar studies in the past (*DeVault et al., 2015*, *2017*, *2018*), we were unable to score alert behavior. Mallards did not exhibit a visible behavior that could be described as alert (raised head, consistent monitoring of vehicle, *etc.*) frequently enough to consistently score.

### *Response variables: distance to vehicle path & movement corrections in field trials*

In the field experiment, we collected a third metric, the perpendicular distance from the road's centerline to the location (to nearest 0.5 m) of the bird at the time of its flight, or the position of the mallard at the hypothetical time of collision if the vehicle was not required to brake (distance to the vehicle path). We analyzed this distance to vehicle path as a dependent variable because a mallard was capable of seeing the vehicle for the entire approach distance (550 m; see discussion of mallard visual acuity in Simulated Trials section), and therefore any movements during the trial could have been related to the approaching vehicle. All movements, both forward and backward in the case of FID correction, and side-to-side, for the distance to vehicle path metric, were measured in post. A 5 × 5 m grid was drawn on the roadway with chalk, with vertical and horizonal lines crossing the grid in 0.5 m intervals. This resulted in FID adjustments and the distance to vehicle path metric having a precision of 0.5 m. The position of an individual was determined based on the center of the body.

### Response variables: presence of flight response, probability of successful avoidance

In both simulated and field arenas, we scored a binary of whether or not a mallard displayed a flight response, as well as whether the mallard would have successfully avoided the vehicle in a real approach scenario (*i.e.*, without implemented safety measures). During some animal-vehicle encounters, birds might not exhibit a flight response (*Blackwell et al., 2012*, *2019*), which was evident in our experiments. A mallard that initiated flight after a collision would have occurred was considered to have no flight response, as it would have been struck by the vehicle before exhibiting flight in a real-world scenario.

In simulated trials, a duck's flight path was limited by the walls of the video box. Therefore, to determine whether a duck would have "succeeded" in avoiding the vehicle, we compared each individual's TTC to the minimum time required for a mallard to escape along an ideal escape path (perpendicular to the vehicle) for 3 m (see *DeVault et al., 2015*), which we determined was 1.0 s (see Simulated Trials, above). As such, any individual that initiated flight less than 1.0 s before collision was deemed unsuccessful (*i.e.*, a virtual collision occurred).

In field trials, we had more information available to determine whether an individual was successful in avoiding a hypothetical vehicle collision than in simulated trials, where we relied only on a TTC threshold of 1.0 s. Realistically, an individual with a TTC indicating a successful avoidance (>1.0 s) in a simulated trial might not actually escape collision in a real-world scenario. For example, a mallard may have a TTC > 1.0, but inadequate subsequent flight speed or trajectory could result in a collision during a real vehicle approach. Similarly, an individual with a TTC < 1.0 s might successfully avoid collision should its path to escape require less than 3 m of movement. As such, we added a second "successful escape" metric in the field, in which a flight was deemed successful if the bird removed itself from the path of the vehicle (1 m on either side of the centerline, given the vehicle's width of 2 m, or vertically clearing the vehicle's height of 2 m) by the time of collision. Birds that initiated flight after braking were considered to have "failed", as presumably none would have removed themselves from the path of the vehicle had braking not occurred.

### General analysis

All analyses were performed using R Statistical Software (Version 2022.07.2+576; *R Core Team, 2022*). All samples were checked for outliers using Grubbs test (*Grubbs, 1969*) from the "outliers" package (*Komsta, 2022*). We evaluated the results of models with the "car" (*Fox & Weisberg, 2019*), "sjstats" (*Lüdecke, 2021*), and "emmeans" (*Lenth, 2022*) R packages. Variation is reported as standard deviation unless otherwise stated. All log transformations used natural log (base 2.718). For all models, we first assessed the effects of sex relative to body mass and found that males were larger than females ($\bar{x}_{Male} = 1.16 \pm 0.031$ kg, $\bar{x}_{Female} = 0.99 \pm 0.015$ kg; $F_{60,1} = 22.57$, $P < 0.001$). Therefore, we chose to use the continuous variable (body mass) rather than sex to maximize degrees of freedom.

### Simulated trial models

We first evaluated the probability mallards displayed a flight response to simulated vehicle approaches in the video lab setting. We used a generalized linear model with a logit link function to evaluate the binary flight response (1) or no flight response (0), with independent variables of vehicle speed, lighting treatment (night *vs*. day; hereafter, time of day), the interaction of speed and time of day, and mallard body mass (log transformed). Vehicle speed and time of day (*i.e*., as categorical variables) were treated as an interaction because the visual stimulus of approaching light is different at night compared to day (*Verheijen, 1985*) and the looming qualities of this stimulus used by birds to determine speed may be altered in different ambient lighting conditions, causing the effective speed (*i.e*., speed that is perceived by a viewer) to vary (*Kim, Perrone & Isler, 2017*).

To evaluate which, if any, margin of safety (spatial, temporal, or delayed) mallards employed, we used two linear models with FID and TTC as respective responses. For the analysis of simulated trials, independent variables were vehicle speed, time of day, the interaction of speed and time of day, and mallard body mass (log transformed). To better meet the assumptions of homogeneity of variance and the normality of residuals, FID was square root-transformed and TTC was log-transformed.

We next explored whether speed or time of day affected successful avoidance among individuals that initiated flight. We fit a generalized linear model with a logit link function to a binary response: all birds with TTC > 1.0 s were considered successful (1), and those with TTC values < 1.0 s were considered unsuccessful (0). Independent variables were vehicle speed, time of day, the interaction of speed and time of day, and body mass (log-transformed).

### Field trial models

Analyses of field experiment data were similar to the simulated experiment, with a few minor differences. First, we included ambient air temperature as an independent variable in models for probability of flight, FID, TTC, and probability of successful escape because of its potential influence on avian FID and overall vigilance (*Møller, 2014*; *Hammer et al., 2022*). As expected, some mallards flew away before the vehicle approach could begin; these birds were not used in any of our analyses. Some individuals exhibited a flight response which occurred after braking when the driver was compelled to stop to avoid collision. Individuals that initiated a flight response after the vehicle braked were omitted from analyses on FID and TTC, as the change in stimulus caused by braking might have affected their avoidance behaviors. To better meet the assumptions of homogeneity of variance and normality of residuals, we log transformed FID and TTC values. We fit a linear model for distance to vehicle path using the same independent variables as those used in the FID and TTC models.

We used a generalized linear model with a logit link function to evaluate the role of vehicle speed, time of day, the interaction of speed and time of day, body mass (log-transformed), and temperature, as well as whether braking occurred post-flight, to account for any additional time which may have been afforded to an escaping individual, on the binary outcome of escape. The responses were successful (1; outside the vehicle's path at

collision) or failed (0; inside the vehicle's path at collision). We also ran a generalized linear model with a logit link function using the metric of successful escape used in simulated trials (TTC > 1.0 s) for comparison, with vehicle speed, time of day, the interaction of speed and time of day, body mass (log-transformed), and temperature as dependent variables, as well as whether braking occurred before flight initiation to account for the confounding factor.

**Ethical Note:** The Institutional Animal Care and Use Committee of the University of Georgia approved all procedures used in this study (A2021 07-001-Y1-A3). A total of 97 adult mallards (30 female, 67 male) was used in this study. Mallards were wild-type (*i.e.*, *A. p. platyrhynchos*) and were acquired from D&D Duck Farm, LLC in Ellerbe, NC, USA. Mallards were housed in a brooder house on the campus of the Savannah River Ecology Laboratory with continuous access to flowing water. Food was provided *ad libitum.* Inherent risks are present when approaching animals with a vehicle. However, these approaches are necessary to learn how animals react in real-world scenarios when the full range of their flight behavior, as well as environmental distraction, are available. We mitigated this risk by using an experienced driver, TLD, who has driven for similar methodologies in the past; we established mandatory braking points corresponding to speed, so that the vehicle could not strike a bird which remained at its release point and monitored the mallard's position in real time using an infrared camera. One mallard was struck indirectly on the windshield at a low speed (<10 km/h) during field trials, as it fled toward the vehicle during braking. It immediately flew away without apparent difficulty, leading us to believe no substantial injury took place. All mallards were released during their respective approach in the field from 14–17 February 2022 on the U.S. Department of Energy Savannah River Site. This was approved by the Savannah River Site in Site Use Permit #SU-21-47-R.

## RESULTS

### Simulated trial results

Mallards were generally calm after placement in the video box and throughout the 5-min acclimation period, but one individual continually attempted to escape the box during acclimation and was excluded from analyses. Of the mallards remaining after the removal of the flighty individual, 48 (50.5%) displayed a flight response. The overall model for the probability of a flight response (*n* = 95) in simulated trials was significant ($\chi^2$ = 19.94, d.f. = 6, *P* = 0.003, log likelihood = −55.87, pseudo $R^2$ = 0.15). The probability of flight was significantly affected by time of day ($\chi^2$ = 10.66, d.f. = 1, 91, *P* = 0.001); mallards were 53.4% less likely to have a flight response during the night (31.6% ± 7.6 SE) than daytime conditions (67.8% ± 7.1 SE). There was no significant effect of approach speed (60 km/h: 55% ± 9.0 SE; 120 km/h: 61.8% ± 9.8 SE; 240 km/h: 32.6% ± 16.5 SE) nor interaction between approach speed and time of day (Table 1).

Across treatments for birds exhibiting flight responses, mean TTC was 2.15 s ± 2.27 SD (range = 0.11–14.73), and mean FID was 64.0 m ± 53.8 SD, (range = 3.5–245.5). One TTC value for an individual in the 60 km/h Day treatment was identified as an outlier by a Grubbs test (TTC = 14.73 s, *P* = 0.01) and with this outlier excluded, mean TTC was 1.88 ±

**Table 1 The effects of independent variables on transformed values of FID and time-to-collision, as well as distance from path of vehicle (field experiment) from general linear models.** $P < 0.05$ bolded.

| Model | F | d.f. | $\omega_p^2$ | P |
|---|---|---|---|---|
| **(sqrt) FID (m)-simulated** | | | | |
| *Approach speed* | 0.76 | 2, 41 | 0.033 | 0.473 |
| *Time of day* | 0.47 | 1, 41 | 0.092 | 0.495 |
| *(log) Body mass* | 0.43 | 1, 41 | −0.017 | 0.515 |
| *Approach speed: time of day* | 1.66 | 2, 41 | 0.027 | 0.203 |
| **(log) TTC (s)-simulated (Outlier included)** | | | | |
| *Approach speed* | 2.41 | 2, 41 | 0.089 | 0.103 |
| *Time of day* | 1.28 | 1, 41 | 0.092 | 0.265 |
| *(log) Body mass* | 0.40 | 1, 41 | −0.014 | 0.529 |
| *Approach speed: time of day* | 0.48 | 2, 41 | −0.022 | 0.622 |
| **(log) TTC (s)-simulated (Outlier removed)** | | | | |
| *Approach speed* | 2.15 | 2, 40 | 0.100 | 0.130 |
| *Time of day* | 5.15 | 1, 40 | 0.194 | **0.029** |
| *(log) Body mass* | 0.94 | 1, 40 | −0.016 | 0.938 |
| *Approach speed: time of day* | 0.79 | 2, 40 | −0.025 | 0.794 |
| **(log) FID (m)-field** | | | | |
| *Approach speed* | 9.81 | 1, 28 | 0.152 | **0.004** |
| *Time of day* | 0.03 | 1, 28 | −0.023 | 0.870 |
| *(log) Body mass* | 0.57 | 1, 28 | −0.018 | 0.457 |
| *Air temperature* | 0.42 | 1, 28 | 0.027 | 0.524 |
| *Approach speed: time of day* | 3.22 | 1, 28 | 0.061 | 0.084 |
| **(log) TTC (s)-field** | | | | |
| *Approach speed* | 11.03 | 1, 28 | 0.235 | **0.003** |
| *Time of day* | 0.03 | 1, 28 | −0.030 | 0.855 |
| *(log) Body mass* | 0.08 | 1, 28 | −0.029 | 0.781 |
| *Air temperature* | 0.42 | 1, 28 | −0.025 | 0.522 |
| *Approach speed: time of day* | 1.81 | 1, 28 | 0.020 | 0.189 |
| **Distance to vehicle path-field** | | | | |
| *Approach speed* | 10.53 | 1, 32 | 0.102 | **0.003** |
| *Time of day* | 0.39 | 1, 32 | −0.026 | 0.537 |
| *(log) Body mass* | 0.44 | 1, 32 | −0.015 | 0.511 |
| *Air temperature* | 1.75 | 1, 32 | 0.019 | 0.195 |
| *Approach speed: time of day* | 6.90 | 1, 32 | 0.134 | **0.013** |

1.32 s (range = 0.11–4.86). For back-transformed mean FID and TTC values across treatment levels, see Table 3.

The overall model for (sqrt) FID in simulated trials ($n = 48$) was not significant ($F_{6,41} = 2.205$, $P = 0.062$, adjusted $R^2 = 0.13$), nor were any independent variables (Table 1). We obtained different results for (log) TTC, depending on the inclusion/exclusion of the outlier we identified. The overall model for (log) TTC with the outlier included ($n = 48$)

**Table 2** The effects of independent variables on probabilities of flight and successful escape (>1.0 s TTC), as well as successful escape (exit from vehicle path; field only) from generalized linear models. $P < 0.05$ bolded.

| Model | $\chi^2$ | d.f. | P |
|---|---|---|---|
| **Probability of flight-simulated** | | | |
| *Approach speed* | 3.43 | 2, 92 | 0.180 |
| *Time of day* | 10.66 | 1, 91 | **0.001** |
| *(log) Body mass* | 0.62 | 1, 90 | 0.432 |
| *Approach speed: time of day* | 5.23 | 2, 88 | 0.073 |
| **Probability of successful escape-simulated** | | | |
| *Approach speed* | 9.65 | 2, 92 | **0.008** |
| *Time of day* | 0.65 | 1, 91 | 0.420 |
| *(log) Body mass* | 0.30 | 1, 90 | 0.585 |
| *Approach speed: time of day* | 2.15 | 2, 88 | 0.342 |
| **Probability of flight-field** | | | |
| *Approach speed* | 2.92 | 1, 60 | 0.087 |
| *Time of day* | 1.18 | 1, 59 | 0.278 |
| *Braking* | 28.88 | 1, 58 | **<0.001** |
| *(log) Body mass* | 0.08 | 1, 57 | 0.775 |
| *Air temperature* | 1.62 | 1, 56 | 0.203 |
| *Approach speed: time of day* | 0.14 | 1, 55 | 0.707 |
| **Probability of successful escape (<1.0 s TTC)-field** | | | |
| *Approach speed* | 7.95 | 1, 60 | **0.005** |
| *Time of day* | 3.39 | 1, 59 | 0.066 |
| *Pre-flight braking* | 1.90 | 1, 58 | 0.168 |
| *(log) Body mass* | 0.58 | 1, 57 | 0.446 |
| *Air temperature* | 0.90 | 1, 56 | 0.343 |
| *Approach speed: time of day* | 2.28 | 1, 55 | 0.131 |
| **Probability of successful escape (Exit vehicle path)-field** | | | |
| *Approach speed* | 0.20 | 1, 60 | 0.653 |
| *Time of day* | 0.92 | 1, 59 | 0.338 |
| *Post-flight braking* | 3.76 | 1, 58 | 0.053 |
| *(log) Body mass* | 1.62 | 1, 57 | 0.203 |
| *Air temperature* | 0.47 | 1, 56 | 0.491 |
| *Approach speed: time of day* | 0.79 | 1, 55 | 0.373 |

was not significant ($F_{6,41} = 2.29$, $P = 0.053$, adjusted $R^2 = 0.14$), and none of the independent variables were significant (Table 1). However, the overall model for (log) TTC with the outlier removed ($n = 47$) was significant ($F_{6,40} = 3.40$, $P = 0.008$, adjusted $R^2 = 0.24$). Time-to-collision in this model was 53.6% higher at night than during the day (Table 1; Fig. 2). Vehicle speed (60 km/h: 1.04 ± 0.10 SE; 120 km/h: 1.122 ± 0.10 SE; 240 km/h: 0.90 ± 0.16 SE) and the interaction between vehicle speed and time of the day were not significant (Table 1).

**Table 3 Back-transformed values for response variables of interest.**

|  | Transformed mean | Mean |
|---|---|---|
| **FID, from sqrt (x + 1)-simulated** | | |
| *Approach speed–60 km/h* | 6.27 | 38.3 m |
| *Approach speed–120 km/h* | 8.38 | 69.2 m |
| *Approach speed–240 km/h* | 10.03 | 99.6 m |
| *Day* | 6.71 | 45.0 m |
| *Night* | 9.74 | 93.9 m |
| **TTC, from log (x + 1)-simulated (Outlier included)** | | |
| *Approach speed–60 km/h* | 1.15 | 2.16 s |
| *Approach speed–120 km/h* | 1.11 | 2.03 s |
| *Approach speed–240 km/h* | 0.92 | 1.51 s |
| *Day* | 0.84 | 1.32 s |
| *Night* | 1.29 | 2.63 s |
| **TTC, from log (x + 1)-simulated (Outlier removed)** | | |
| *Approach speed–60 km/h* | 1.04 | 1.83 s |
| *Approach speed–120 km/h* | 1.12 | 2.06 s |
| *Approach speed –240 km/h* | 0.90 | 1.46 s |
| *Day* | **0.77** | **1.16 s** |
| *Night* | **1.28** | **2.60 s** |
| **FID, from log (x + 1)-field** | | |
| *Approach speed –40 km/h* | **4.43** | **82.9 m** |
| *Approach speed–60 km/h* | **3.43** | **29.9 m** |
| *Day* | 3.63 | 36.7 m |
| *Night* | 4.23 | 67.7 m |
| **TTC, from log (x + 1)-field** | | |
| *Approach speed–40 km/h* | **2.19** | **7.94 s** |
| *Approach speed–60 km/h* | **1.18** | **2.25 s** |
| *Day* | 1.53 | 3.62 s |
| *Night* | 1.84 | 5.30 s |

**Note:**
Marginal mean values of flight initiation distance (FID) & time-to-collision (TTC) back-transformed from square-root (sqrt) and natural log (log) transformations for two treatments of interest, approach speed and time of day. Bold values indicate statistically significant comparisons (within models).

The overall model of the probability of successful avoidance in simulated trials ($n$ = 95) was significant ($\chi^2$ = 12.75, d.f. = 6, $P$ = 0.047, log likelihood = −55.58, pseudo $R^2$ = 0.10). The probability of a successful avoidance was significantly affected by vehicle speed (Table 2); mallards were 228.9% more likely to successfully avoid the vehicle at 120 km/h than at 240 km/h ($z$ = 2.78, $P$ = 0.015), although success at 60 km/h did not differ significantly from either 120 km/h ($z$ = −0.62, $P$ = 0.811), or 240 km/h ($z$ = 2.26, $P$ = 0.061; Fig. 3). The probability of a successful avoidance did not vary significantly with time of day (Day: 37.9% ± 7.6 SE; Night: 29.7% ± 7.3 SE) or the interaction between vehicle speed and time of day (Table 2).

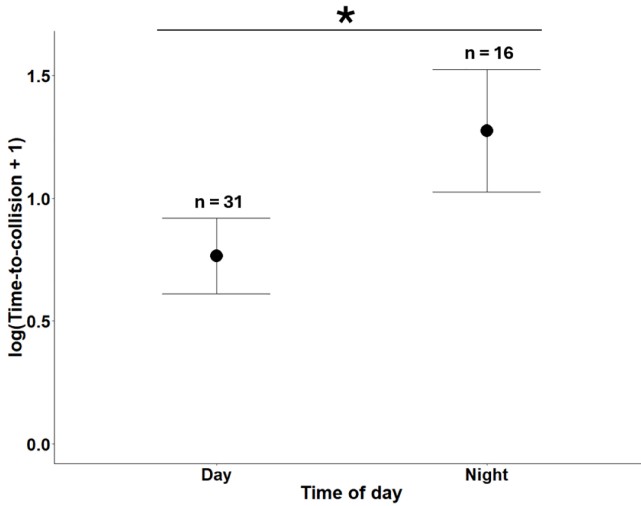

**Figure 2 Marginal means of log(Time-to-collision + 1) (s) with 95% confidence intervals among mallards exposed to a simulated vehicle approach filmed at different times of day.** A significant effect, indicated by asterisk, was found when an outlier in the 60 km/h Day treatment was excluded.

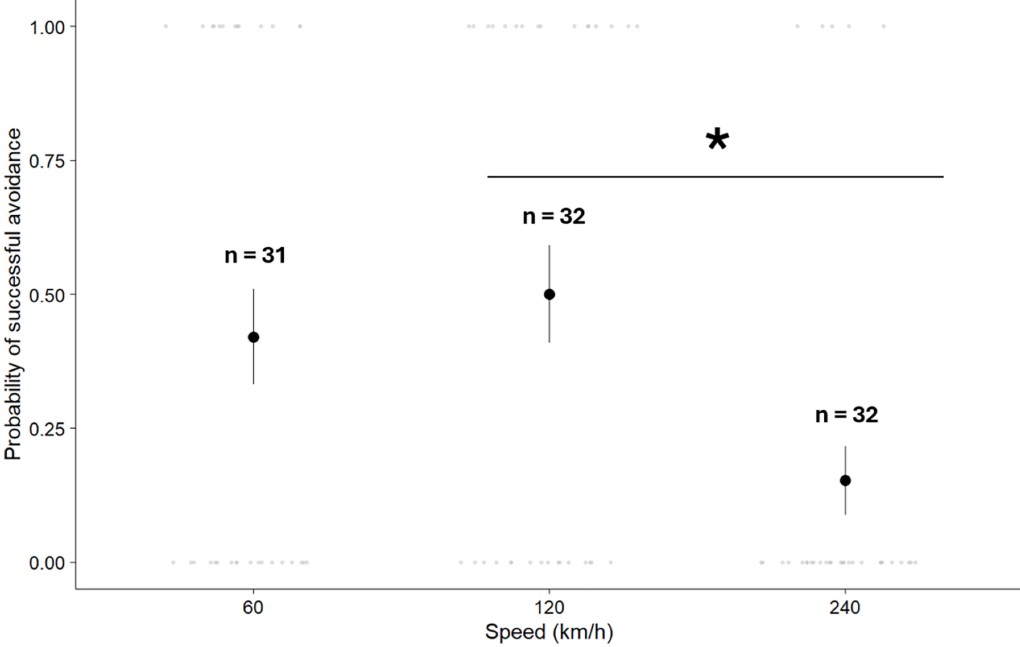

**Figure 3 Marginal means and 1.5 interquartile ranges of a mallard's probability of successful avoidance (time-to-collision > 1.0 s) at three experimental speeds during simulated vehicle approaches.** A significant difference, indicated by asterisk, was found between 120 & 240 km/h speeds.

## Field trial results

Some mallards flew away immediately upon release. Sixty-two of 96 total birds (64.6%) remained on the road after release long enough to be scored. Of the 34 mallards that flew

before the approach, 18 (52.9%) were females, which comprised 60.0% of our total female mallards.

Of the scorable birds, 41 (66.1%) displayed a flight response, although seven of the 41 initiated flight after braking and were not included in general linear models for FID, TTC, and distance to vehicle path. Four mallards that did not have a flight response had moved themselves far enough from the vehicle's path to not require braking and were included in the model for distance to vehicle path. Across treatments for birds exhibiting flight responses ($n = 41$), mean TTC was 7.03 s ± 9.57 SD (range = 0–34.02). Mean FID was 84.1 m ± 111.3 SD, (range = 0–378). For back-transformed mean FID and TTC values across treatment levels, see Table 3.

The overall model for probability of flight response in the field ($n = 62$) was significant ($\chi^2 = 34.82$, df = 6, $P < 0.001$, log likelihood = −22.28, pseudo $R^2 = 0.44$) and indicated that mallards which required braking initiated flight at a lower rate (21.2% ± 9.3 SE) than those that did not (94.9% ± 3.7 SE; Table 2). Seventeen of 21 mallards that did not show a flight response required braking; however, it is unlikely that those mallards opted not to fly *because* they were braked for, given that braking occurred so late in the vehicle approach. Vehicle approach speed was not significant despite the 44.7% nominal decrease in flight responses as speed increased (40 km/h: 85.0 ± 8.6%; 60 km/h: 47.0 ± 13.6%; Table 2). Time of day (Day: 70.0% ± 13.4 SE; Night: 68.4% ± 16.3 SE) as well as the interaction between approach speed and time of day were not significant (Table 2).

The overall model for (log) FID in the field ($n = 34$) yielded non-significant results ($F_{5,28} = 2.20$, $P = 0.080$, adjusted $R^2 = 0.16$). However, when examining the individual effects, vehicle speed was significant (Table 1), by which mean mallard (log) FIDs at 40 km/h were 29.2% longer compared to 60 km/h (Fig. 4A). Time of day (Day: 3.63 ± 0.40 SE; Night: 4.23 ± 0.44 SE) as well as the interaction between approach speed and time of the day were not significant (Table 1). The significant effect of speed should be interpreted cautiously given that the overall model was not significantly better than a null model based on the F-ratio test. Although speed does have a significant effect on FID, there are likely other influential untested variables as indicated by the overall model's $R^2$.

The overall model of (log) TTC was significant ($n = 34$; $F_{5,28} = 2.69$, $P = 0.041$, adjusted $R^2 = 0.20$). Vehicle speed was significant, with mallards initiating flight with 85.6% more time before collision at 40 km/h than 60 km/h (Fig. 4B). Time of day (Day: 1.53 ± 0.32 SE; Night: 1.84 ± 0.35 SE) and the interaction between vehicle speed and time of day were not significant (Table 1).

The overall model quantifying mallard distance to the vehicle path at flight or vehicle arrival ($n = 38$) was significant ($F_{5,32} = 2.89$, $P = 0.029$, adjusted $R^2 = 0.20$). Vehicle speed was significant (Table 1), with mallards staying 59.0% farther from the vehicle path at 60 km/h (1.24 m ± 0.17 SE) than at 40 km/h (0.78 m ± 0.18 SE). Time of the day was not significant (Table 1; Day: 1.19 m ± 0.21 SE; Night: 0.83 m ± 0.23 SE). We found a significant interaction between vehicle speed and time of day (Table 1; Fig. 4C), whereby the distance to vehicle path did not differ significantly between speeds at night (40 km/h: 0.92 m ± 0.29 SE; 60 km/h: 0.74 m ± 0.29 SE; t = 0.50, d.f. = 32, $P = 0.618$) but was
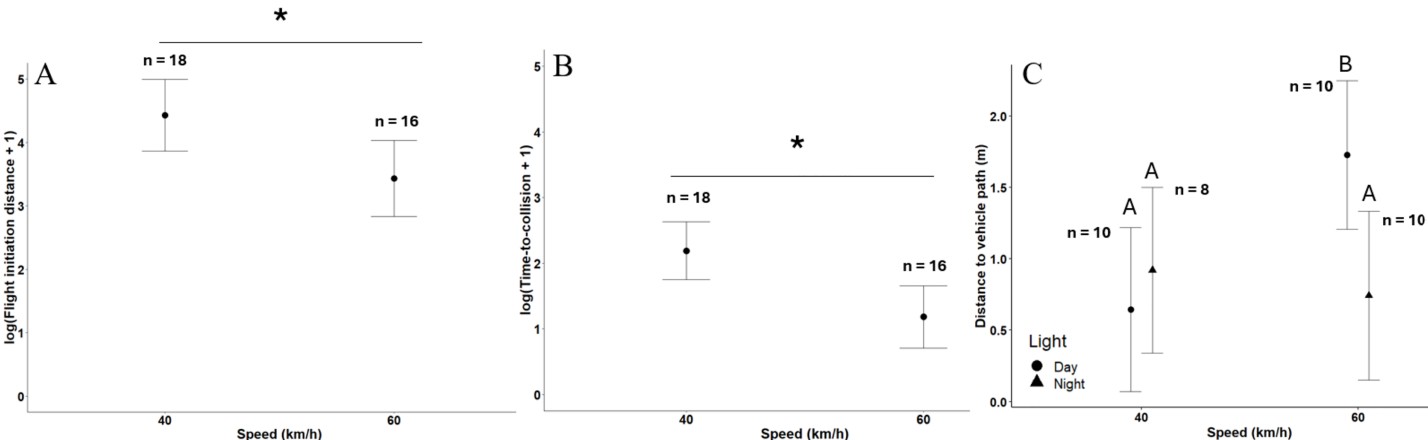

**Figure 4 Effects of vehicle speed (km/h) during field approaches on (A) log(flight initiation distance + 1), (B) log(time-to-collision + 1), and (C) the effect of the interaction between vehicle speed and time of day on distance from vehicle path.** Error bars represent 95% confidence intervals. Significance is indicated by an asterisk (*) in (A) and (B). In (C), significance is indicated by lettering (A, no significant difference; B, significant difference).

significantly shorter during the day at 40 km/h (0.64 m ± 0.28 SE) than at 60 km/h (1.73 m ± 0.26 SE; t = −3.25, d.f. = 32, $P$ = 0.003).

The overall model quantifying successful avoidance of the vehicle in the field, as determined by whether the individual remained in the path of the vehicle at the time of hypothetical collision ($n$ = 62), was not significant ($\chi^2$ = 7.76, d.f. = 6, $P$ = 0.256, log likelihood = −35.81, pseudo $R^2$ = 0.10), nor were any of the independent variables (Table 2). However, the overall model analyzing successful avoidance using the theoretical success metric used in simulated trials (successful avoidance requires >1.0 s TTC; $n$ = 62) was significant ($\chi^2$ = 17.00, d.f. = 6, $P$ = 0.009, log likelihood = −34.44, pseudo $R^2$ = 0.20). The probability of successful avoidance was significantly affected by vehicle speed (Table 2; Fig. S3), by which mallards were more than three times less successful at avoiding the vehicle at 60 km/h (14.6% ± 8.6 SE) than at 40 km/h (55.8% ± 12.9 SE). Time of day (Day: 29.5% ± 13.7 SE; Night: 34% ± 16.6 SE) and the interaction between vehicle speed and time of day were not significant (Table 2).

## DISCUSSION

This study is the first to explore avian reactions to oncoming vehicles at night, when bird strikes are more likely (*Dolbeer, 2006*) and are especially concerning during migratory periods (*Dolbeer et al., 2023*). Roadways, and thus terrestrial vehicles, also pose additional risks for *Anseriformes* at night, especially during the breeding season (*La Sorte et al., 2022*). We found that mallards approached by real vehicles displayed lower FIDs as vehicle approach speed increased. Furthermore, we found that nighttime conditions during simulated approaches reduced the probability mallards would respond to a vehicle approach, but when they reacted, they did so with more time to spare until a potential collision occurred than under daytime conditions.

Mallards tested in the field experiment had decreased FIDs and TTCs at high *vs.* low speeds, indicating a delayed margin of safety. This result was not observed in simulated

approaches. However, no *a priori* power analysis was conducted before data collection. Because of this, we cannot decisively conclude that non-significant results are representative of true mallard responses. It is possible we lacked a sufficient sample size to detect all significant results. Notably, the delayed margin of safety was only observed when individuals were tested at relatively low speeds in the field (40 and 60 km/h), and the result did not differ between day and night. We found that the overall model for FID in the field was non-significant. While we conclude that vehicle speed does affect FID, any number of unconsidered variables account for a large proportion of observed variance in mallard responses. These variables may include, but are not limited to, eye direction at release, percent time monitoring vehicle approach, mallard age, or solar irradiance. In a largely unaltered outdoor arena, there also exist any number of factors which can introduce statistical noise (*i.e.*, moving leaves and branches, insects on the roadway, bird songs, *etc.*).

Although a spatial margin of safety has been observed in other bird species (*Cárdenas et al., 2005*; *DeVault et al., 2015*), this is the first study to observe a delayed margin of safety in a bird. A delayed margin of safety can result from distracted monitoring of a potential threat (*Lunn et al., 2022*). Unlike the simulated experiment in which the mallards were allowed 5 min to acclimate to the arena, field approaches began immediately upon their release to reduce the probability of the mallard escaping prior to approach. This method gave them little time to take in their surroundings after being held in a dark environment and provided potentially distracting visual stimuli (*i.e.*, the novel environment of an open roadway) which might have reduced their assessment time of the approaching vehicle. Any distraction from monitoring an oncoming threat allows more time to elapse before a flight response, thus allowing a faster vehicle to approach more closely before evoking a flight response.

Alternative to the distraction hypothesis, the daytime increase in distance to vehicle path–when the body of the vehicle is most visible–at the higher speed might not indicate distraction, as suggested by *Lunn et al.*'s *(2022)* theoretical model of the delayed margin of safety, but difficulty processing potential risk at the higher speed. Specifically, the period of low-quality assessment (time between object detection and alert response) and high-quality assessment (time between alert and flight; *Tyrrell & Fernández-Juricic, 2015*) could decrease with increasing approach speed (*DeVault et al., 2015*). There are also responses other than flight which birds could employ when attempting to escape a predator, such as hiding in place (*Sordahl, 1982*; *Lima, 1993*). If mallards perceived they were obscured or not in the direct path of an approaching threat, remaining in place might be a preferred antipredator response. It should be noted that neither experimental arena represented a mallard's typical habitat, and their typical predator escape strategies of diving underwater or flushing into herbaceous vegetation (*Lima, 1993*) were not possible. Perhaps a delayed margin of safety was representative of a period of confusion given a relatively contrived scenario for an obligate waterbird.

During the simulated trials, mallards were less likely to exhibit avoidance behavior in reaction to nighttime videos. Without accompanying audio cues, the video stimulus might have been perceived as less threatening than in a real setting, despite previous research suggesting mallards do not typically rely on auditory cues for threat assessment (*Conomy*

et al., 1998). The immobility shown by many of the mallards during nighttime trials also could reflect a period of assessment (DeVault et al., 2015), or the nighttime atmosphere could have given the mallards more confidence they could hide in place. Because the effective speed of a looming object illuminated by bright light can be harder to discern in dark settings (Kim, Perrone & Isler, 2017), it is also possible the looming stimulus would not be seen as threatening until it is too late to react to avoid collision (Blackwell et al., 2019, 2020). Although mallards were less likely to respond to the nighttime stimulus, when they did take flight, they did so with more time before a potential collision than in response to the daytime stimulus. This response could result from differences between individuals' risk thresholds. Intraspecific variability in avian fear responses to humans (Carrete & Tella, 2011) could also be present as a response to anthropogenic stimuli like vehicles. For example, differing magnitudes of response to disturbance can arise depending on the type of threat approaching, even between vehicle types (Hardy & Crooks, 2011; McLeod et al., 2013).

In simulated trials, we saw no conclusive evidence of mallards using any defined escape strategy, as speed affected neither FID nor TTC. During simulated approaches, the vehicle could have been approaching too quickly (up to 240 km/h) for the mallards to use defined antipredator strategies, a phenomenon that was observed in turkey vultures (DeVault et al., 2014) and brown-headed cowbirds (DeVault et al., 2015). What seems clear from the simulated experiment, however, is that ambient lighting during simulated approaches can influence both the probability and timing of mallard flight responses.

Time of day also affected field approaches. We found that mallards located themselves farther from the vehicle's path of travel at the time of flight or collision at the higher speed, but only during the day. The birds positioned themselves farther from the vehicle when the visual stimulus loomed more quickly in daytime conditions. This could indicate the mallards perceived the vehicle as something other than a predation threat (Lunn et al., 2022); rather, their perception of risk depended on the directness of the vehicle's approach to their position, a response previously found in other bird species (Wang & Frost, 1992; Møller & Tryjanowski, 2014; Lima et al., 2015) during which flight occurs at longer distances when the bird is approached directly by a threat. Critically, though, time of day had no bearing on mallards' observed margin of safety–if birds reacted, they had similar mean FIDs and TTCs regardless of ambient lighting condition.

The large proportion of mallards that did not exhibit any reaction to the approaching vehicle was unexpected (50.5% in simulated approaches; 33.9% in field approaches). Non-flights in response to vehicle approaches have been observed in previous vehicle-approach experiments with some bird species (Blackwell et al., 2012, 2019; DeVault et al., 2017), but not in others (DeVault et al., 2014). Mallards used in this study were raised on a farm and we received them at ages of 3–6 months. Given their life history, it is plausible that the simulated vehicle observed in our experiments was their first experience with the visual stimulus of any vehicle. In a review of predator neophobia, birds, captive-raised animals, and animals with a trophic role lower than tertiary consumers displayed significantly higher levels of neophobia than other taxa, wild-caught animals, and predators (Crane & Ferrari, 2017). The mallards used in this experiment possess

all three of these qualities, although the lack of flight behaviors in sizeable proportions of our population indicate these mallards may not have been particularly susceptible to neophobia. Although birds that do not flee when approached by a vehicle are inherently at a higher risk of collision, probability of flight does not fully encompass an avian-vehicle interaction. Other metrics, like the probability of collision, should also be considered.

In both simulated and field experiments, the probability of successful avoidance (>1.0 s TTC) was lower at high speeds. Low TTC values at high speeds seem reasonable given the temporally shorter approach at higher speeds and delayed margin of safety observed in the field. During field approaches, the theoretical probability of successful avoidance decreased as speed increased from 40 km/h (55.8%) to 60 km/h (14.6%). This apparent sensitivity to relatively small changes in speed is relevant given that modern cars on highways and aircraft on runways are traveling many times faster than our experimental field vehicle. Notably, whether a bird would have actually avoided a collision (*i.e.*, the binary metric of success defined by a mallard's exit from the vehicle's path) in the field was not affected by speed or any other variables, demonstrating a need to consider additional metrics in animal-vehicle collision studies, beyond the time of flight, to determine whether a collision would occur (*Blackwell et al., 2020*).

Flight initiation distance and time-to-collision values are only one component of an animal's total flight response, along with the direction and angle of flight, the sustained velocity of the flight, and the distance needed to clear the vehicle's path (*Blackwell et al., 2019*). Our results indicate that at higher vehicle speeds, mallards initiate avoidance responses with very little time available to avoid collision (*sensu Bernhardt et al., 2010*). Although we cannot be certain whether an escape attempt would truly be successful in an actual wildlife-vehicle encounter, any TTC < 1.0 s results in a hazardous scenario. This is especially true if a late reaction by an animal causes the vehicle operator to swerve, or when there are additional forces acting on the area around the vehicle, like the intake of a jet engine. Lastly, the assumption used in this study (and in *DeVault et al., 2015*, *2017*), that a 3 m flight is needed to avoid collision, is a useful tool for analyzing factors contributing to a hazardous encounter. However, this assumption is not always conclusive in terms of whether a collision would occur. In simulated vehicle encounters, however, it remains the best available proxy, as the ability to analyze subsequent flight characteristics is limited.

## CONCLUSIONS

As previously observed in vultures (*DeVault et al., 2014*), brown-headed cowbirds (*DeVault et al., 2015*), and mourning doves (*Blackwell et al., 2009a*), our results indicate mallards will often fail to avoid vehicles when they approach at the takeoff speed of most aircraft (~240 km/h). Notably, mallards are the first avian species to be observed employing a delayed margin of safety which, if present in real-world vehicle approaches, would make vehicle encounters with these birds especially hazardous. To reduce collisions, the presence of waterbirds around airfields should be discouraged (*Blackwell et al., 2009b*; *DeVault et al., 2011*). Separation in space between these birds and vehicles is necessary because the faster the vehicle, the more likely a collision is to occur. However, it is often impractical to completely remove wildlife from these areas, especially in the case of birds

like mallards, which thrive in human-dominated environments (*Figley & VanDruff, 1982*). Recognizing that a complete separation in space between mallards and aircraft is challenging, our results provide a few, more moderate suggestions. Firstly, the riskiest time to encounter a mallard is at night, when they may be less likely to initiate avoidance behavior and are less likely to change their position relative to a vehicle at high speeds. Mallards are frequently active at night (*Korner et al., 2016*), especially during migration, when they will fly long distances (>200 km) over the course of one night (*McDuie et al., 2019*). We suggest, then, that during the months of migration, aircraft flight should be minimized during the night and at the altitudes used most by migrating ducks as much as is practical. Although mallards have been struck by aircraft at altitudes up to 6,400 m (*Manville, 1963*), most individuals migrate at altitudes less than 915 m (*Lincoln & Peterson, 1979*). Of the 636 mallard strikes for which elevation is reported (March 2024), 598 (94.0%) occurred below 1,000 m (*United States Federal Aviation Administration, 2024*). An approximate "danger zone" for mallards therefore could be described as ground level to 1,000 m. Based on our findings here, we recommend future research should focus on improving the visual saliency of high-speed vehicles to birds (*Blackwell et al., 2012*; *Goller et al., 2018*), to increase the probability birds will detect and avoid oncoming aircraft sooner, before collisions are imminent. It is also important to determine which other species may use a delayed margin or safety in response to vehicle approaches, and under what conditions this response manifests. Animals using a delayed margin of safety are inherently hazardous when approached at high speeds, but the mechanisms responsible for this behavior are not yet identified.

## ACKNOWLEDGEMENTS

We thank B. Blackwell, J. Martin, and T. Sasaski for helpful review of manuscript drafts, and J. Hoblet for field assistance.

### Funding

Funding support was provided by the U.S. Department of Energy Office of Environmental Management under Award Number DE-EM0005228 to the University of Georgia Research Foundation and the Warnell School of Forestry and Natural Resources at the University of Georgia. There was no additional external funding received for this study. The funders had no role in study design, data collection and analysis, decision to publish, or preparation of the manuscript.

### Grant Disclosures

The following grant information was disclosed by the authors:
U.S. Department of Energy Office of Environmental Management: DE-EM0005228.
University of Georgia Research Foundation.
Warnell School of Forestry & Natural Resources, University of Georgia.

## Competing Interests

The authors declare that they have no competing interests.

## Author Contributions

- Shane Guenin conceived and designed the experiments, performed the experiments, analyzed the data, prepared figures and/or tables, authored or reviewed drafts of the article, and approved the final draft.
- Carson J. Pakula performed the experiments, authored or reviewed drafts of the article, and approved the final draft.
- Jonathon Skaggs performed the experiments, authored or reviewed drafts of the article, and approved the final draft.
- Esteban Fernández-Juricic analyzed the data, authored or reviewed drafts of the article, and approved the final draft.
- Travis L. DeVault conceived and designed the experiments, performed the experiments, authored or reviewed drafts of the article, secured funding, and approved the final draft.

## Animal Ethics

The following information was supplied relating to ethical approvals (*i.e.*, approving body and any reference numbers):

The University of Georgia IACUC provided approval for this research (Approval number: A2021 07-001-Y1-A3).

## Field Study Permissions

The following information was supplied relating to field study approvals (*i.e.*, approving body and any reference numbers):

Field experiments were approved by the U.S. Department of Energy-Savannah River Site under a site use permit (Approval number: SU-21-47-R).

## Data Availability

The field data and simulated data, metadata describing the column headings in the .csvs, and R Code used for data analysis and figure generation are available in the Supplemental Files.

These files are also available at Open Science Framework: Guenin, Shane. 2024. "Inefficacy of Mallard Flight Responses to Approaching Vehicles." OSF. August 30. doi:10.17605/OSF.IO/K9Y8M.

## Supplemental Information

Supplemental information for this article can be found online at http://dx.doi.org/10.7717/peerj.18124#supplemental-information.

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
