# Peer review of "Inefficacy of mallard flight responses to approaching vehicles"

_PeerJ, doi:10.7717/peerj.18124_

## Round 0.1 · original submission · Minor Revisions

Overview

This manuscript reports the results of two experiments examining various components of the avoidance response of mallard ducks to an approaching vehicle at different speeds in the day and night. The stimulus was a pickup truck with airplane landing lights to mimic the effect of a plane on a runway. In the simulation, ducks were placed in a box with opaque walls except for one that was mesh facing a television screen use to play a recording of the truck approaching at speeds of 60, 120 and 240 km/h. In the field trials, the ducks were released on a road, and the same truck approached at speeds of 40 and 60 km/h. In the field some individuals were not assessed because they fled before the stimulus was presented. The authors recorded whether or not the mallards fled, if so, the FID, TTC, and whether the duck reacted in time to have avoided being struck. In both studies the sample size was substantially reduced for measurements specific to the fleeing response because only about half the individuals fled. In the simulation, the probability of fleeing was not affected by speed but was higher during the day; in the field, there was no effect of speed or time on the probability of fleeing. In the simulation, FID was not affected by speed or time of day; in the field, FID surprisingly was shorter at the higher speed. In the simulation, TTC was not affected by speed but was longer at night; in the field, TTC was lower at the higher speed, but was not affected by time. In the simulation, the estimated probability of avoiding a collision was lowest at the highest speed (but not significantly different from the lowest speed) and not affected by time; in the field, probability was lowest at the highest speed and not affected by time. The authors emphasize the novelty of the ‘delayed margin of safety’ in the field which both FID and TTC were reduced at higher speed and the different responses at night.

Both reviewers found the manuscript generally sound in both analysis and presentation and made several recommendations for changes, although Reviewer 1 requested clarification that individuals were not subjected to repeated trials with different treatments. I agree with Reviewer 1 that a more direct comparison of the two experiments is warranted, especially for recommendations for future research and concerns about the reliability of video presentations. I also feel that some important elements of the methods need to be added and that potential internal contradictions within each experiment should be addressed. For my comments below, you can treat them as a third review, i.e., make changes if they are valid and provide a detailed explanation if they are not. Numerous small grammatical suggestions and notes of typos have been made on the attached pdf using highlights and inserted comments. You do not have to mention these in your response unless you disagree with the suggestions.

Major concerns

The ‘delayed margin of safety’ in the field experiment is the primarily emphasized finding. Essentially, this is the result of a lower FID at higher speed. If FID decreases with speed, TTC must also decrease, whereas if TTC decreases, FID could have decreased, stayed constant or even increased to a small extent, depending on the speed difference. This finding needs to be discussed more thoroughly. How robust is the result, given the lack of statistical significance of the overall model? What is the magnitude of the effect? Could it be an artifact of the study design? (If a mallard is released on the road and a truck starts toward it right away, a fixed time for the animal to assess the context, recognize a threat, and begin to flee would result in a lower FID for a faster moving truck. Is there anything in the video which would provide insights into this possibility?) I recognize that there is a brief mention of this in the Discussion, but I feel that it needs a more complete and critical evaluation. Are there other possible explanations for the effect? You state that this is the first bird species to show this response but do not mention whether it has been found in studies of other taxa, although it is implied by specifying ‘bird’. In presenting the literature, previous evidence for a lower FID at higher speeds in any species and to any type of stimulus would be highly relevant.

The use of a nighttime approach is presented as one of the knowledge gaps that the study addresses, but I did not find an adequate synthesis of how things are different between night and day. There are multiple non-significant effects and some significant ones and differences between the simulation and field study, but they need to be integrated to identify what, if anything, this study actually contributes to our understanding of this topic. Again, some implications are suggested, but there is not a rigorous discussion of the evidence.

I am not expert in the statistical approaches used but wonder whether the sample size is adequate for the conclusions. There are multiple treatments, a sample size reduced by individuals that did not flee or that fled before the treatment started, and quite a large number of predictor variables. Please check with an independent statistical expert to assess whether your approach is appropriate and to what extent lack of significant difference could be due to a true lack of effect rather than to issues of variability, sample size, and number of independent variables. Could this explain contradictory patterns such as a lack of effect of speed on FID but no decrease in TTC in the simulation? You could also ask this advisor how you should treat the FID result where the overall model is not significant but you want to draw conclusions about the effect of speed.

There should be a more synthetic comparison between the simulation and the field study. The inconsistency of many results should be recognized and possible explanations, including but not limited to statistical issues mentioned above, addressed. The noise of the approaching truck, the presence of the release person, the range of speeds tested, recognition of the video all might be important. A more rigorous examination of the potential limitations of both studies and the implications for future research on this topic would be very helpful.

The title highlights the inefficacy of the response. I feel this is an important point which should be the focus of a separate paragraph in the Discussion. This conclusion must be nuanced by consideration of the validity and limitations of both experiments. How much more inefficacious is the mallard response compared to other vehicle responses and to approaches by people and real predators? The present Abstract is not strongly supportive of this perspective: “. . .suggest mallards might be . . .’. See my specific comments below for a few thoughts on how the Discussion might be organized.

Other suggestions

Title: The title is somewhat ambiguous. How does ‘make way’ fit into the studied patterns?

Abstract.
L15. To make it easier for readers, please keep a consistent order of presenting the simulation and field experiment through the abstract, methods, and results.
L21. ‘Time to collision at reaction’ is an awkward expression and may not be clear to readers. Try to find a more direct way to express this.
L22-25. This sentence is very vague and does not make it clear whether this explanation is offered for the lower probability of reaction or the greater TTC.

Introduction
L61. I suppose vehicles may also lack specific cues by which animals recognize their predators?
L70. TTC is an important concept throughout the manuscript and deserves a fuller explanation. ‘Collision’ seems to be a concept for vehicle approach, but the framework here is more general. Is ‘collision’ used in predatory contexts? Clarify that it is not the time to the actual collision but to the estimated time to contact if no evasive action is taken.
L74-79. Is it important to also indicate what threat was being responded to in these cases?
L80. Is delayed margin of safety appropriately included in a list of strategies? It seems hard to think of a case where it could be effective, so is it really a strategy?
L80. Is distraction the only potential source of such an effect? Any process resulting in a fixed reaction time (lifting head, focusing eyes, processing information, generating flight) could result in a decrease in both FID and TTC with speed, and this would be particularly apparent at high speeds.
L93. Is there a null hypothesis? What is expected if the animal makes no adjustment to changes in approach speed? Is there any value here in clarifying that FID can increase but that TTC will still decrease if the increase is not enough to compensate for the speed? This would be in a category of responses to speed that do not provide adequate protection, especially at high speeds.
L94. I had to check what you meant by ‘salient’ in this context. Your use is correct but I wonder if there an alternative word (conspicuous? prominent?) that might be more readily recognized by an international audience?
L111. Here or in a previous paragraph it would be appropriate to discuss the strengths and limitations of the two approaches and the history of their use for vehicle collision studies.

Methods
L116. Although mallards are so widespread that most scientists will probably know them, it would still be appropriate to provide a bit of background: that it is a duck, its size, abundance, tameness, tameness around humans, use of terrestrial, aquatic and aerial habitats, . . .
L153. Were the headlights and aircraft lights lighted both day and night?
L171. I assume you mean that each successive individual was subject to a different, randomly selected treatment, not that you did all of one treatment before going on to then next treatment. Perhaps rewrite to make this clearer.
L172, 225. Was the capture truly random (e.g., by a random number table) or haphazard? As Reviewer 1 notes, there is evidence for bias in capture order for a range of animal species. However, I don’t see a problem with this as long as the order of treatments was not associated with the order of capture. If there was such a correlation, it needs to be carefully discussed as it could be a serious problem.
L174. Important to clarify which were the dimensions of the box facing the screen.
L176. It seems likely that the screen did not fill the entire space in front of the box. Give us a better idea of what the mallard’s view was and what they might see around the sides of the screen.
L180. So, it seems that the day and night treatments were both performed during the day and under the same lighting conditions. Is this likely to have affected the mallards’ perception or responsiveness? Consider making this more explicit here and also whether it is a topic for discussion when you consider the validity of the findings.
L205.What is the point here? If the ducks fled into the trees, it seems to imply that the wooded edge could have had an effect.
L217. Were the lights on both day and night?
L221. How long, what distance/time to accelerate to the designated speeds?
L236. Missing information: starting point of the truck (in Fig. 1 but should be in text also), whether the truck drove down the center of the road (affects significance of distance from center measure), how were the birds released, what determined when the truck started moving forward, where the observer who released the duck was positioned, and whether this observer moved after releasing the duck, possibly affecting the duck’s movement.
L238. Missing information: explicit statement of precisely what movement determined the time of the start of the mallard’s escape (since we are dealing with a time scale of seconds, this could be critical), how the distance of the truck was assessed, whether distance and time were estimated independently or from the relationship in Fig. 1 caption, whether the truck was assumed to be traveling at the experimental speed even if it was still accelerating, the units of distance and time (without this information, log(x + 1) cannot be calculated. On L396, there is an indication that mallards could move out of the path without a flight indicating that the definition of ‘flight’ is quite important.
L239. I can’t see the correction for duck movement in Fig. 1. The specific correction is important because of the angles involved.
L241. Was the duck’s position the center of the body or the closest part of the body?
Fig. 1. This figure has a problem with perspective. It is not clear if it is a view from above (as implied by the distance to path) or from the side (as implied by lateral views of duck and truck). ‘Flight’ position needs to be more explicit that this is the position of the truck at the time the mallard flees (perhaps insert a second, lighter figure of the truck at this point?). Specify that the solid lines at the top and bottom of the figure represent the edges of the road. It is not critical, of course, but the figure would be more useful if the truck looked more like the pickup used and the duck looked more like a mallard than a cartoon.

Results
L355. If a TTC value is an outlier, shouldn’t the associated FID also be an outlier? Is it valid to exclude an individual from TTC analysis but include the same individual in the FID analysis?
L356. Shouldn’t you also indicate which treatment the outlier occurred in?
L363. I find ‘53.4% less likely’ rather awkward; could it be clearer to write ‘2.1 times more likely to have a flight response during the day than during the night’? [If you decide to change it here, use equivalent expressions elsewhere in the Results.]
L370. To me, it is unnecessary to give all your Results with the outlier included and excluded, although it is helpful for the overall means in the first paragraph. Check with a statistician about this, but it seems that if you identified it as a statistical outlier and state that you excluded it, it would be reasonable to provide only the results without it (relevant to text, table and figure).
L374. You cited Fig. 2b without citing Fig. 2a.
Table 1. In the heading, identify all abbreviations, including column heads. You wrote out time to collision but abbreviated FID. Check whether you are consistent in use of hyphens for time to collision throughout manuscript, including figures. Check whether df normally has periods. (Apply similar corrections to Table 2.)
Fig.2. I found it quite difficult to interpret the transformed values. Even though you need to transform for analysis, I think readers would find it much clearer to have actual values than having to interpret (TTC + 1), especially without being certain whether the pre-transformation units are seconds. Specify the significance of the asterisk. I would find it more useful to have panel A show FID (untransformed) instead of the pattern with the outlier. Indicate the sample size for each treatment, for example above the points. All figures have quite small numbers and axis labels. Make a copy of each figure and reduce it to the size it would likely appear in the printed version and check that the numbers and letters are easily visible. You may have to use a larger font.
Fig. 3. Also, identify all abbreviations in this figure. I don’t understand putting the dots at zero and one as if it was a logistic regression. They do not correspond to different speeds, and the mean provides the indication of how many there are at 0 and 1. In fact, I wonder if a bar graph with proportion responding and proportion not responding would be clearer. An indication of the sample size for each of the three treatments is important.
Figures 2 and 3 overall. I think that a single figure with all 6 treatments would be a lot clearer. You could still show significant different by brackets above the points.
L389-392. Wouldn’t it be more appropriate to provide this information after presenting the number that fled (L398)?
L391. I don’t think readers should have to go to supplementary information for something as fundamental as means and variations for each treatment for each variable. Please try to find a way to incorporate into a table or figure (using back transformed values, not logs).
L414. This is a rather wishy washy statement. I recognize that it complicated, due to the effect of the overall model (which is nearly significant). However, it is an important element of one of your main conclusions so you need a clearer statement about the level to which you accept this finding. You should consult a statistician about how much to trust this finding and how best to convey it to readers.
Fig. 4. I think presenting both speeds and times for all three variables in untransformed units would be clearest here. Placing the key outside the box unnecessarily widens the figure forcing a smaller size of the main information in the printed version; I suggest describing the key in the caption or putting the key inside one of the panels. Define the error bars in the caption. Provide the sample sizes for each treatment (as numbers above the bars, perhaps?).

Discussion
Organization: I found the Discussion somewhat challenging in terms of clarity and logical flow of ideas. However, neither reviewer mentioned this, so it may be just me. It would be a good idea to check with other readers to see if this is an issue with others. Remember that for each conclusion you need to consider how strong and consistent the evidence is and then how it relates to previous literature. A smoother and more logical flow might be as follows: Start with the low response rate and address possible reasons and whether it creates concerns about the conclusions based on those individuals that did respond. Compare to literature on response rate to vehicles and to other stimuli such as people and predators. Then focus on each of your experimental treatments, speed and time of day. Starting with speed, consider its effect on FID, TTC, and avoidance success in both experiments, then do the same with time. It would be useful to briefly present how an animal might recognize differences in approach speed. Since nocturnal tests are one of your original contributions and one on which you base recommendations, a critical evaluation of what the evidence shows is particularly important. Again, what are the potential effects of light level on the recognition of an approaching threat. Finally, you could have a paragraph addressing implications for reducing wildlife collisions followed finally by one addressing implications for testing methods.
L443-446. This is like the Introduction and not needed here.
L446-453. While it is helpful to readers to present a brief overview of the results at the start of the Discussion, some of this material is not actual findings but conclusions based on those findings. Interpretations must be preceded by a careful analysis of the strengths and weakness of the evidence.
L463. Be more explicit about what the stimuli were, how they could have been distracting, how this would lower FID, and how it would reduce the effect of speed on FID.
L465. The topic sentence implies you are addressing distance to the path but the conclusion of the paragraph implies that you are providing an explanation for the delayed margin. Very confusing.
L467-469. Not clear why a shorter assessment implies difficulty in assessment or how this would affect the effect of speed.
L473. I think you can present the decision to freeze vs. flee without invoking the mallard’s emotions.
L475. As they are dabbling ducks, I wonder if diving is truly a normal escape behavior.
L476. I didn’t find this very clear about how confusion would result in a reduction in FID with speed.
L478. This paragraph needs to be revised to make the logic of your suggestions more explicit. How precisely could the suggested issues contribute to lower likelihood of response and increased TTC at night? You seem to be referring to two effects but not making it clear which explanation goes with which effect.
L511. Explain why this is critical. Is there a potential contradiction between the distance and FID, TTC effects?
L514. I did not see the 35.3% in the Results and could not see how it was calculated. Please be explicit in the Results.
L515. The magnitude of non-responses in previous studies is important as is how it compares to approaches with other stimuli.
L524. Your conclusion needs to be more explicit.
L533. Is a 50% increase in speed really ‘relatively small’?
L535. Clarify the apparent contradiction between this line and L531-532.
L537. Elaborate on what these might be and why FID and TTC are insufficient.
L569. This suggestion assumes similar reactions between a sedentary duck and a duck in flight already moving. Is this assumption reasonable?

References
Although I did not check everything, your reference section has the fewest errors I have observed in a long time! Congratulations and thank you. The only mistake I noticed is that Behavioral Ecology uses American, not British spelling (perhaps more easily recognized as I was once the editor!)

Supplementary Table S1. Replace kph with SI units km/h
Typo in the heading for the data set (‘of’ not ‘or’)

Reviewer 1 ·

Basic reporting

I first would like to commend the authors for the amount of work achieved. This manuscript was generally easy to read and well reference, and the introduction did a good job at explaining the current hypotheses that exist with regard to FIDs, something I was not familiar with. Figures were self-explanatory as well.

Few comments:

Intro
line 35. The citations (DeVault et al., 2014, 2015) are not sufficient here. For such a general statement, you should really cite reviews or papers that cover multiple species/taxa as well as modes of transportation.
l36. 'animal responses to vehicles are poorly understood': Citation? I can think of thorough review by Lima et al. (2015)
l58. Two consecutive sentences start with 'However'.
l74-75. On the topic of a temporal margin of safety: may be relevant to discuss here or in discussion, Legagneux & Ducatez (2013) [10.1098/rsbl.2013.0417] show that avian species may adjust their FIDs to the speed limit rather than the actual speed of oncoming vehicles
l87-88. You have left some parentheses that do not make sense
l104-114. I am missing elements on one of the models here: why did you measure and investigate the position of the duck relative to vehicle's path center?

M&M is overall pretty long and confusing when describing statistical analyses. Suggest summarizing the different response variables that were measured, either at the beginning or end, or adding sub-titles to paragraphs.
l124-125. Were the birds also raised without/with limited exposure to vehicles?
l170. 'each treatment video was played for 16 unique individuals'. Do you mean 16 ducks in total, each shown 6 videos in random order, or 16 different birds captured for each video?
I initially understood the former, which does not track with the number of data points on line 360 (n=95, suggesting 16 ducks * 6 videos minus 1 flighty duck, and not 16 - 1 ducks * 6 videos) and suggest your all statistical analyses for the simulated trials are invalid, because the observations are not independent. It would also raise questions regarding the validity of the simulated trials, which may be biased if the 16 birds captured were the most adventurous (easiest to approach and capture). Therefore, I believe you meant the latter, which merits rephrasing to improve clarity.
l205: 'given that many ducks chose to flee into the cover of the trees' citation? I believe personal observations from the author(s) would suffice here.
l239. 'corrected for duck movement after release when necessary in the field'. I did not understand this sentence, do you mean FID and TTC are calculated from the final duck location rather than release location? It would seems a pretty obvious thing to do.

Results
l353. I would have liked to know right away what percentage of birds did not flee, to know on how many birds the TTC and FID measures rely rather than looking for it in the next paragraph
l394. You have made this point several times already in the methods, I do not believe it is necessary to repeat this here
l400. In the MM, I do not find mention that the variable "required braking" is included the model, nor why it was.

Experimental design

Designs appeared well-researched and sound. It is clear that many steps were put in place to maximize the safety of the birds throughout the study.
I understand the rationale behind the use of simulated trials, but you do not appear to really compare results between the two methods (which share the treatment of 60km/h) to assess the validity of simulated trials in mallards, which could have been useful to other researchers; and helped explain why your results differ between the two approaches.

Validity of the findings

The study does show that mallards are probably not efficient in avoiding oncoming vehicles, and provides some support for the presence of a delayed margin of safety in this species. The authors do a good job of listing potential confounding factors in the discussion.

Statistical analyses seem sound to me, although potential concerns may arise depending on how many subjects were used for the simulated trials (see my concerns in section 'Basic reporting', l170), which I believe is just a matter of phrasing. I see no glaring issues in the raw data nor the R scripts that were provided. Only question I have regarding the stats: could you state why the body mass was log-transformed in the models?

A few concerns regarding the validity of the results:
- How were daytime and nighttime handled during simulated trials? You only state that the boxes had artificial lights (l152-168), were they also on during nighttime videos?

- When did the experiment to assess the success of the escape (l188-198) occur? Was it close to the simulated or field trials? I was wondering about possible confounding effects of repeated experiments on ducks, which may increase levels of anxiety, affecting behaviour.

- How random is the random capture of individuals in the enclosure: was an individual randomly selected and then captured, or were the 'easiest' ducks captured first? In particular, in the 'hypothetical outcome of avoidance' experiment, is there a possibility that the sample (n=20) is biased towards less active birds, artificially decreasing the threshold value?

- Your use of a 1s threshold to discriminate successful/unsuccessful escapes is not ideal, as you transform a measured value (mean +- s.d.) into a hard threshold. Suggest trying to run the models with each extreme (1-0.14, 1+0.14) as a threshold, or better yet, predict the hypothetical success of the escape by randomly drawing from the observed distribution of escape times. This would show whether the results are robust with respect to this experimentally estimated value.

Additional comments

no comment - well done!

Reviewer 2 ·

Basic reporting

• 1. Basic Reporting
Clear, unambiguous, professional English language used throughout. YES
Intro & background to show context. YES
Literature well referenced & relevant. YES, but see comment below
Structure conforms to PeerJ standards, discipline norm, or improved for clarity. YES
Figures are relevant, high quality, well labelled & described. YES
Raw data supplied (see PeerJ policy) I didn’t check the file.

Experimental design

• 2. Experimental design
Original primary research within Scope of the journal. YES
Research question well defined, relevant & meaningful. YES
It is stated how the research fills an identified knowledge gap. YES
Rigorous investigation performed to a high technical & ethical standard. YES
Methods described with sufficient detail & information to replicate. YES

Validity of the findings

• 3. Validity of the findings
The findings are valid and their shortcomings discussed.

No discussion of impact, as directed.

Additional comments

• 4. General comments
My driveway exits onto a long and flat country road with an extended view in both directions. I am very familiar with this particular situation and can judge the distance to an approaching vehicle - but it is difficult to judge its speed, especially at night when all that can be seen are headlights. I have learned to wait for any oncoming vehicle to pass by before pulling out (because the speed is often much higher than the posted speed limit of 60kph).
A paper by Legagneux and Ducatez (European birds adjust their flight initiation distance to road speed limits. Biology letters, 2013) suggests birds do something similar. They found that birds approached by a vehicle based escape decisions on the usual speed on that stretch of road, and not on the actual speed of the vehicle. Consequently, the flight initiation distance (FID) decreased with speed, as also found in this MS. The paper suggests that individual birds are not adept at judging vehicle speed and (like me when leaving my driveway) assign a set speed to oncoming traffic, and cannot fine-tune a response. I am surprised that this relevant paper is not cited, as the literature coverage is otherwise excellent.
The authors might check this paper to confirm the claims they make about firsts achieved by their paper (e.g. line 443). Also, the use of language regarding strategies and tactics can be tricky, and they should carefully assess what they intend to convey. The word strategy is used sparingly (5 times in their MS). Lines 446 – 448 state:
Our results indicate an escape strategy which could be described as a delayed margin of safety (FID and TTC decrease as speed increases) in response to vehicle approach.
Do they mean that the margin of safety is intentionally reduced in high-speed situations, or that this is the result of a perceptual limit imposed by the situation (i.e., unable to judge the speed)? It could be a good strategy to use a set FID when the approach speed cannot be discerned, akin to the error-minimizing threshold of a signal-to-noise problem. These possible interpretations have implications for managing this issue (line 43).

---

## Round 0.2 · accepted · Accept

The authors have provided a detailed and thoughtful response to comments by the reviewers and editor. I consider the paper ready for publication.